# Health Care Financing and Economic Performance during the Coronavirus Pandemic, the War in Ukraine and the Energy Transition Attempt

**Kornelia Piech**

Department of Quantitative Economics, Collegium of Economic Analysis, Warsaw School of Economics, al. Niepodległości 128, 02-554 Warszawa, Poland; nel.piech@gmail.com

**Abstract:** On account of the current epidemiological situation in the world, which results from propagation of the coronavirus, research related to the impact of health on economic growth is becoming especially important. Health capital is an important variable in economic growth models. The method of financing health care has a significant impact on both the health of the population and the level of income. Therefore, this article attempts to analyze the relationship between these values. The way in which health care is financed has a significant impact both on the state of health of the population and on the level of financial resources allocated to health care (e.g., health contributions according to income). The analysis covered the period 2000–2021. On the basis of official reports, available literature and own studies, health expenditure has been divided into three income groups: low-income countries, middle-income countries and high-income countries. On the basis of statistical and economic analyses, it has been found that GDP (Gross Domestic Product) has an impact on public health expenditure in upper- and middle-income groups, but not in low-income countries. The analysis presented is becoming more important in the context of the coronavirus pandemic, the war in Ukraine and energy expenditure related to health care.

**Keywords:** health; economics; financial resources; sustainable development

## 1. Introduction

Since the beginning of the 1990s, many possible determinants to growth have been verified, but only some of them have been included in the canon commonly considered to be the most important [1,2]. The role of human capital in this context is unquestionable [3,4]. A higher level of education, health, new opportunities for learning and improving qualifications are mentioned as the most important aspects related to human capital, which in turn plays an important role from the perspective of economic growth [5,6]. The provision of a minimum level of education and basic health resources is essential for sustainable economic growth [7,8].

Empirical observations on the formation of variables describing human capital suggested an increase in the investment rate, which accompanies economic development, along with an increase in the level of education and health [9,10]. It was noticed that low levels of human capital may constitute a significant limitation on the economic development opportunities of countries, while reducing their competitiveness [11,12]. Therefore, in order to reduce the development gap between poor and wealthy countries, it is necessary to understand some mechanisms responsible for the formation of human capital [13–15].

For a long time, however, the subject literature has ignored health as an important factor affecting economic growth, while primarily identifying the variables describing education with human capital [16,17]. However, basic economic intuition, supported by preliminary empirical research, suggests that health should play an important role in economic growth [18–20].

The first attempts to show the impact of health on economic growth were made by Grossman [21] and Muurinen [22]. In the following years, many scientists tried to find an answer to the question of how health influences economic growth. In their seminal article, Mankiw, Romer and Weil [23] highlighted the need to extend the analysis so as to include the impact of health and nutrition on the human capital factor. The influence of health on economic growth has become a widely accepted and important subject of research for economists. The analyses of Fogel [24], Bloom and Canning [25], Barro and Sal-i-Martin [26] and Barro [27] were among the first works on this subject, which set the direction for further research.

Staying healthy is crucial to the well-being of individuals, which translates into their economic situation [28,29]. Along with good health, the level of human capital increases, which in turn has a positive impact on the individual's productivity and economic growth [30,31]. Good health conditions increase the overall productivity of the labor force, as they are accompanied by lower rates of sickness-induced unavailability or weakness, and fewer days of sick leave [32,33]. They also increase the individual's chance of finding better-paid work [34]. In addition, a high level of health capital allows for a gradual improvement in the level of education, while both extending the period that an individual can spend on learning and increasing the overall level of education through a greater allocation of resources in this direction [35,36].

In order to understand the accumulation of human health capital and wealth, it is essential to be aware of the cause-and-effect relationship between these two streams of values [37,38]. The main difficulty here is to accept the possibility of endogeneity between health and wealth [39,40]. On the one hand, a high level of health capital as a form of human capital may positively affect the productivity of the labor force; on the other hand, higher income also positively affects health capital [41,42]. The ability to achieve higher income results in an increase in the consumption of goods and services that improve health, such as better-quality food, medicine or medical consultations [43]. It should be noted that there are also side effects of certain factors on the health of individuals, such as lifestyle changes, greater involvement in the workplace and higher education levels, which in turn promote better health through increased income [44,45]. The nature of these phenomena may cause bias and inconsistency in the estimators of structural parameters of models that describe the relationship between health and economic growth [46,47].

The answer to the question about the impact of health on economic growth strongly depends on the adopted model assumptions [27]. The most important element of Barro's analysis was to see the two-way cause-and-effect relationship between health and economic growth in its extension of the neoclassical growth model. The model also takes into account the direct impact of health on productivity. The improved health of the population can affect economic growth, while economic growth further accumulates health capital. Meltzer [48] noted that an increase in health capital may lower the rate of depreciation on human capital, which leads to an increase in the rate of return-on-investment in education (thus encouraging investment in education). By contrast, in his book, Howitt [49] identifies six different channels through which health can influence economic growth in its extension of the Schumpeterian view of growth. These include: labor productivity, life expectancy, learning opportunities, creativity, propensity to survive and social inequalities [50].

Zhang et al. [51] attempted to show that economic entities with a higher life expectancy are more inclined to save, and their savings, in turn, support capital accumulation and thus economic growth. In addition, there are convincing microeconomic analyzes in the subject literature which show that these entities will invest more in education, which may also positively affect growth, as demonstrated by Miguel and Kremer [52] and Jayachandran and Lleras-Muney [53]. When the child mortality rate is low, parents are more likely to have fewer children and the fertility rate decreases, which slows down population growth and positively influences the increase in GDP per capita [54]. Therefore, it appears justified to expect that the productivity of healthier economic entities will be higher, that they will

be more creative, or more willing to adapt to technology and rapid changes related to the progressive changes in environmental conditions [55].

In the current economic literature, the answer to the question about the impact of health on economic growth is still ambiguous. On the one hand, there is emphasis on the need to extend the production function in models to include health capital [56]. On the other hand, Acemoglu and Johnson [57], based on empirical analyzes, prove that life expectancy may have an adverse effect on economic growth, opposite to that expected. The COVID-19 pandemic and the war in Ukraine are forcing a new perspective on these issues [58–60].

## 2. Materials and Methods

As health is an important factor in economic growth models, this article examines causality, i.e., the direction of the relationship between health and economic growth. First, the most recent data from the World Health Organization (WHO) were depicted, followed by a Granger causality study between the variables that describe health expenditure and GDP. The manuscript also notes the effects of the COVID-19 pandemic and the Ukrainian war, on economic factors.

### 2.1. Health Expenditure by Cross-Sections

The distribution of health expenditure in the world is very diverse. Data on health expenditure in different countries are analyzed below. The main source of data is from the WHO. The most recent data possible was used.

The highest ratios of health financing in relation to GDP occur in North America (over 10% of GDP, over 13% in United States of America and Cuba) and in most western European countries—especially its "core", i.e., France, Germany, Benelux (excluding Luxembourg), Switzerland, Austria, Sweden, Denmark, as well as Japan and New Zealand. Moreover, several other European and African countries are characterized by a high value of this factor. The lowest rates of overall health expenditure in relation to GDP can be observed in some countries in Central and Southeast Asia. The distribution of health expenditure in the world is very diverse.

### 2.2. Overall Health Expenditure as a Percentage of GDP

This is obviously a relative measure, which shows the emphasis that each country places on health rather than absolute expenditure. In this respect, the geographic distribution is quite similar, i.e., the highest health expenditure is recorded in the US and Canada, in the Scandinavian countries (except Finland), Germany, Austria and Australia. Generally, they are clearly high in the most developed countries. The lowest, in turn, can be found in Central Africa and Myanmar (formerly Burma), i.e., in countries considered to be the poorest.

### 2.3. Health Expenditure per Capita (According to Average Exchange Rates against the US Dollar)

In countries with high health expenditure per capita, public funding for overall health expenditure remains high. According to the WHO, among private expenses on health, 41% were direct, out-of-pocket expenses in 2021. On the other hand, the expenditure on health in total accounted for more than 10% of GDP in 2021 (global average).

### 2.4. The Structure of Expenditure on Health in the World

The highest share of government spending on health in total is recorded in North America and the 'core' countries of the European Union (especially in Switzerland and Belgium) as well as in Japan, New Zealand, Uruguay and Costa Rica. The lowest priority of financing the health system is generally observed in Africa and South Asia.

The breakdown of countries in terms of proportions of funding general health expenditure between private (PHE) and government (GHE) sources, per person, is shown in the chart below (Figure 1). It shows that there are generally all kinds of countries in the

world and there are no major densities in proportion between them. The highest values of private expenditure reach 86% of total expenditure on health, which then corresponds to 14% of the share of government expenditure. However, there are countries where private spending is close to zero—all financing is then covered from their government funds [61].

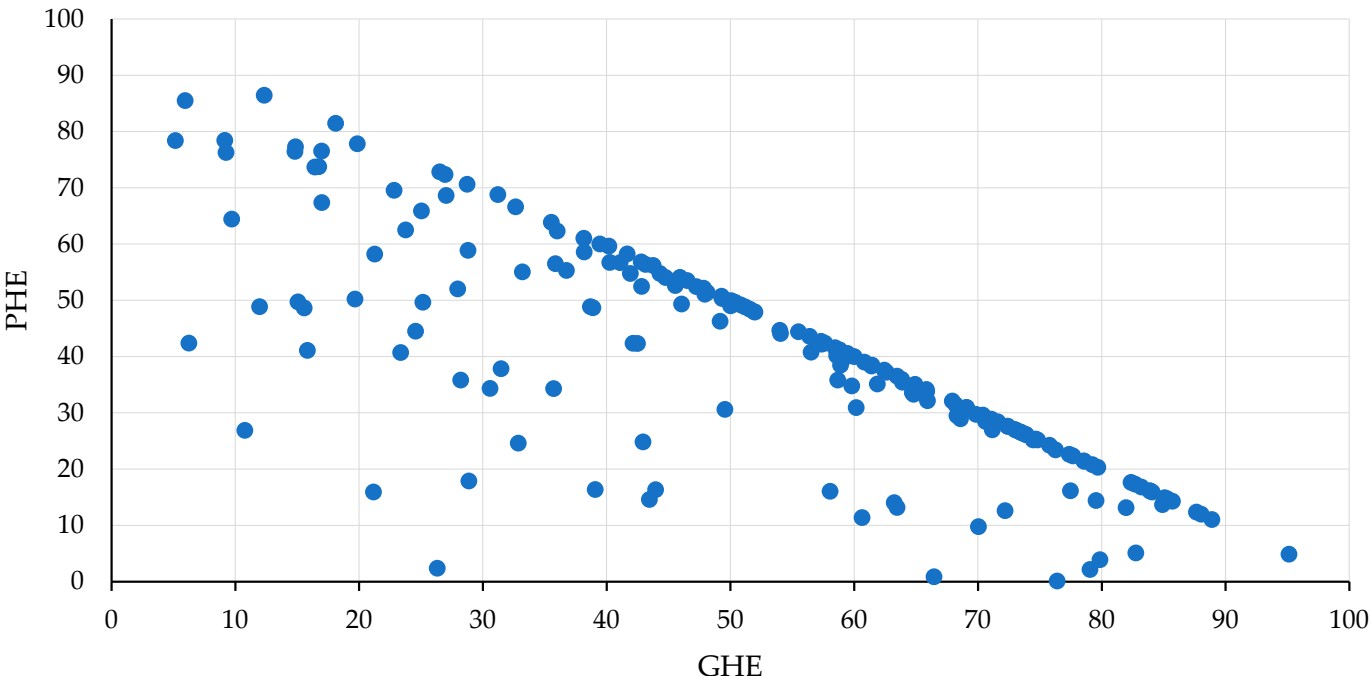

**Figure 1.** Private and government spending on health as a percentage of total health expenditure in 2021, in 189 countries [61].

The most recent available data are presented in the above sections. It may be interesting to compare how health expenditure has changed over time. As can be seen, these expenditures show an upward trend, with government expenditures growing slower. Global spending increased from $600 per person in 2000 to $1459 per person in 2021, and government spending from $326 to $845, respectively (Figure 2) [62].

The differences between the countries and their groups are very large: in high-income countries, the expenditure per capita in 2021 was $6665, and in low-income countries it was only $109 (Figure 3).

However, these proportions are improving. Between 2000 and 2021, low-income countries increased their health spending by 1.33 times, upper-middle-income countries—5.1 times, and high-income countries—2.45 times.

Government spending on health is also increasing: from $1534 in 2000 to $3869 in 2021 in high-income countries, and from only $17 in 2000 to $22.6 in 2021 in low-income countries (Figure 4). Between 2000 and 2021, low-income countries increased their health spending 2.2 times, upper-middle income countries 3.9 times and high-income countries 2.4 times. In poorer countries, the increase in government spending on health per person was higher than the increase in health spending per person in total [63].

In low-income countries, the relationship between public and private expenditure over the entire studied period remains below 64%, while in high-income countries it is above 146% (Figure 5). When observing the changes within the group of upper-middle income countries, it seems that the hallmark of development may be the increasing proportion between government and private health expenditure (per person) [63,64].

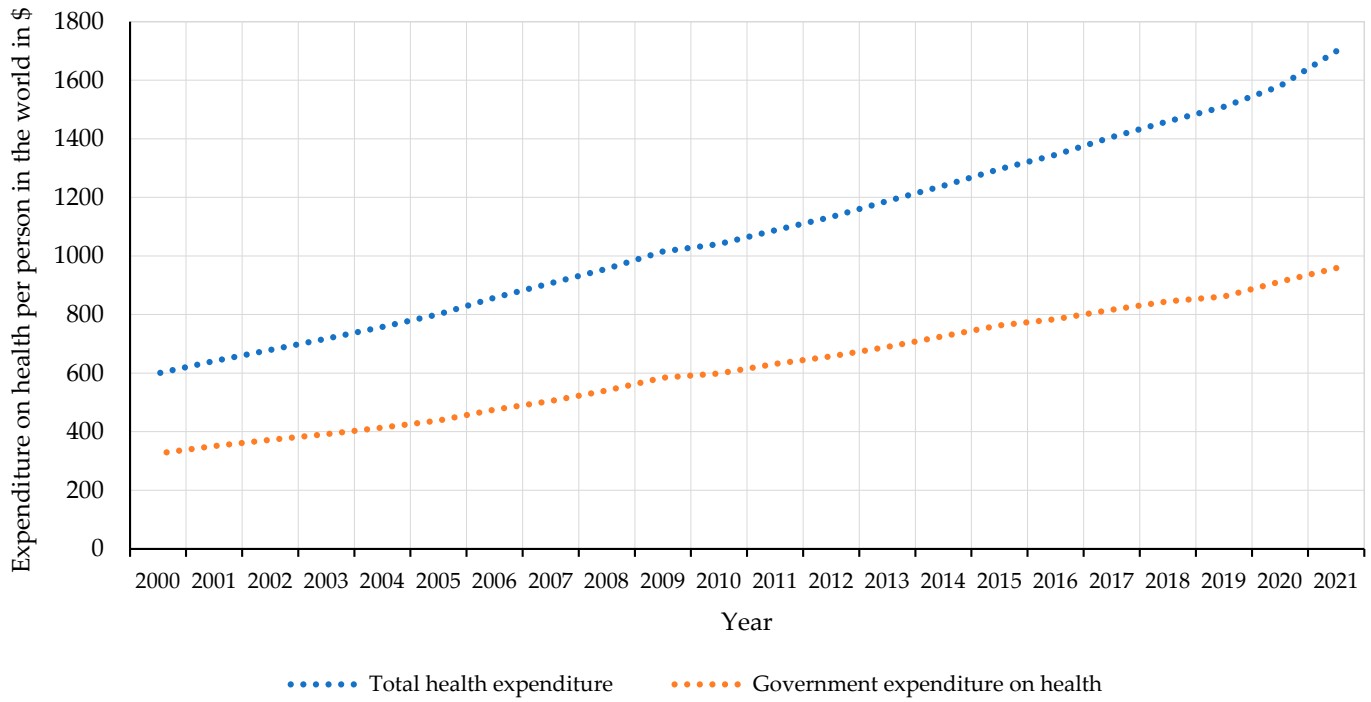

**Figure 2.** Expenditure on health per person in the world, including government expenditure (in international dollars according to purchasing power parity) [62].

In 2000, low-income countries spent only about 2% per capita of what high-income countries did. At present, it is about 1.8%. The proportions of government expenditure changed in a similar way; they decreased from approx. 1.1% to approx. 0.6%, respectively.

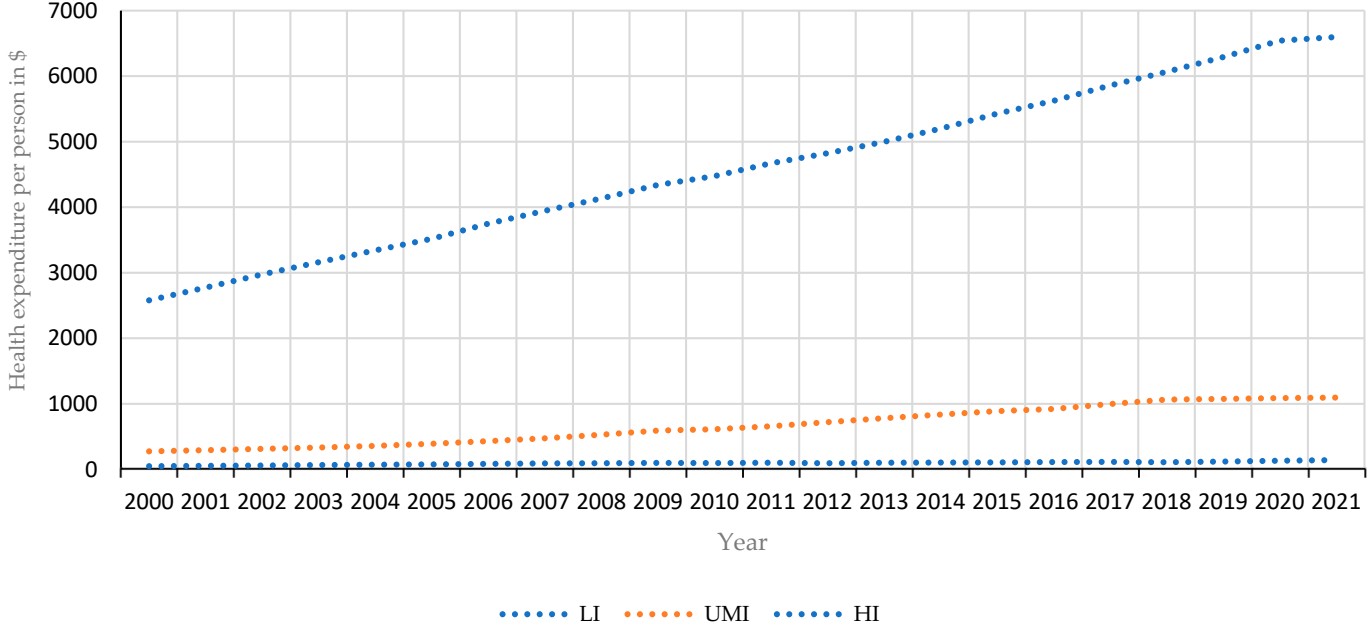

**Figure 3.** Health expenditure per person by income group (in international dollars according to purchasing power parity: LI—low-income countries, UMI—upper-middle income countries, HI—high-income countries [63].

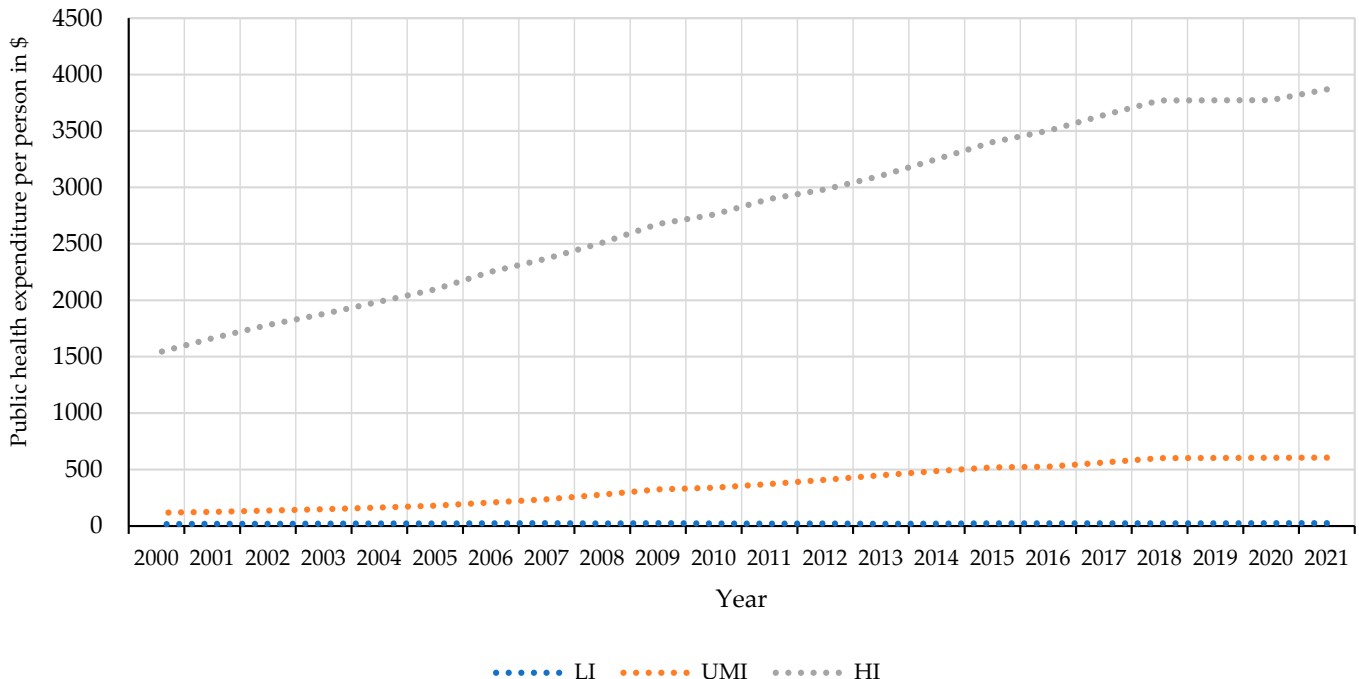

**Figure 4.** Public health expenditure per person by income group (in international dollars according to purchasing power parity) [63].

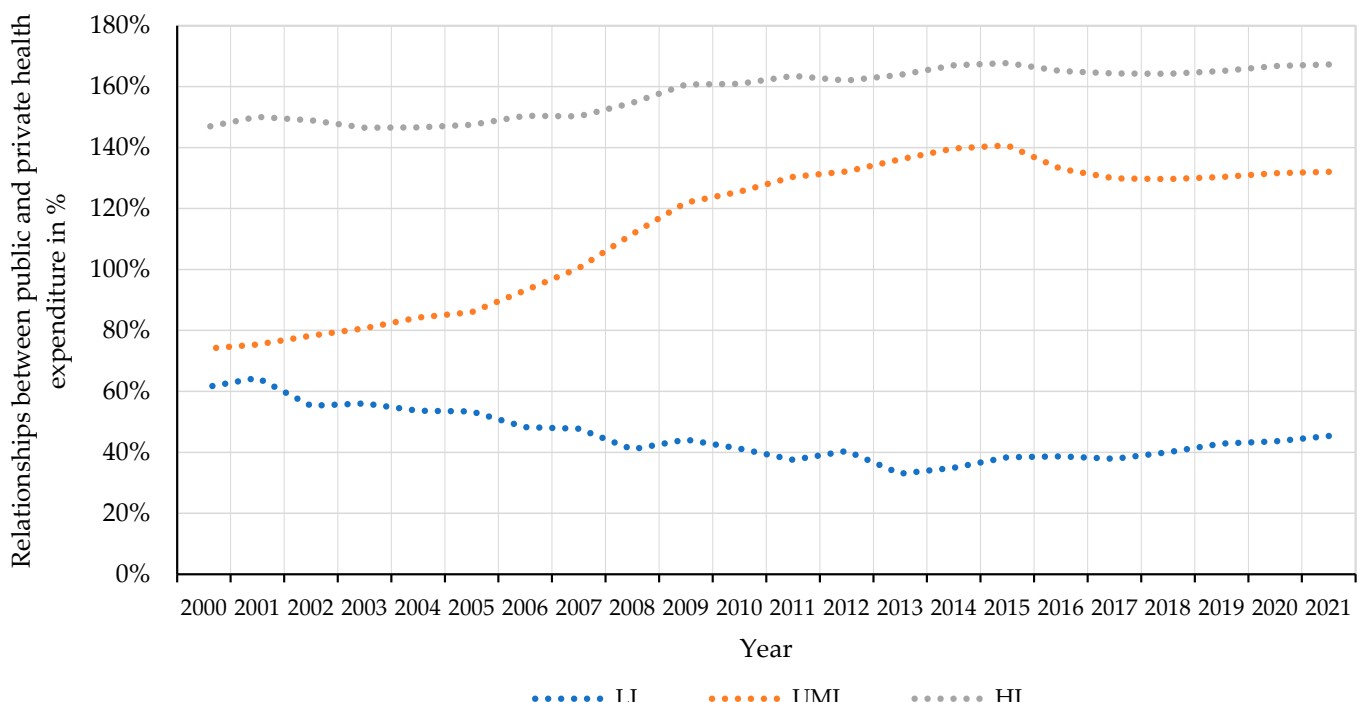

**Figure 5.** Relationships between public and private health expenditure per person in different groups of countries by their income [63,64].

Changes in the difference in levels of private spending can be calculated. They did not change as significantly as in the case of public expenditure. These proportions decreased from 2.6% in 2000 to 2.5% in 2021. This means that—according to the author's calculations—the majority of the process of divergence on health expenditure per person (between 'poor' and 'rich' countries) in 2000–2021 was due to changes in public spending (Figure 6) [65].

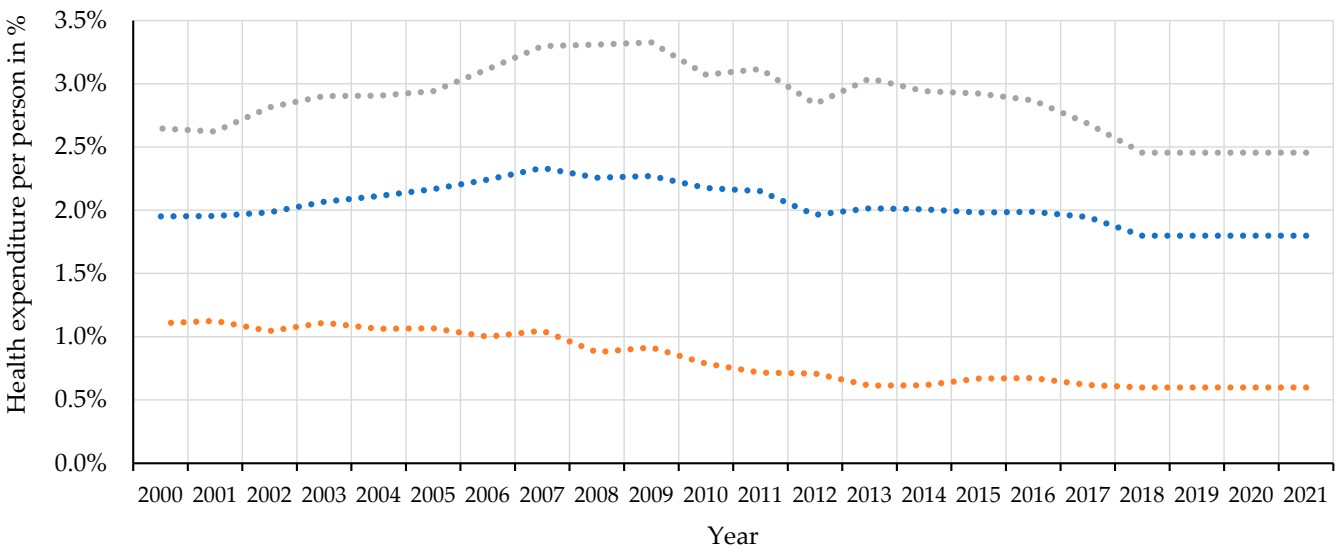

**Figure 6.** Proportion of different categories of health expenditure per person in low-income countries against high-income countries [65].

## 3. Results

### *3.1. Health Expenditure in the World in Three Groups of Countries by Their Level of GDP*

It can be noticed that an increase in GDP is accompanied by an increase in total expenditure on health per capita in all three groups of countries analyzed above. In 2009, more developed countries generally experienced a recession, but despite the decline in GDP, the level of health expenditure per capita continued to increase over the studied period. In low-income countries, there were two periods of decline in health expenditure: in 2010 and in 2012 (Figure 7) [61,63].

A comparison of the values of government and private expenditure per capita in the three groups of countries, according to their GDP levels, showed similar trends. In general, both indicators were accompanied by increases with breaks in the corresponding periods (Figure 8) [61,63].

Such clear trends, as observed above, did not occur in the case of comparing the GDP level to the relationship between government and private expenditure on health per capita in the analyzed groups of countries. In low-income countries, there was a rather sharp increase in government versus private spending in 2000, and then the scale of these relationships fluctuated. In the upper-middle income countries, the ratio between government and private spending was initially observed to decline, but after a period of stabilization this ratio has increased since 2006, along with the increase in GDP.

In the case of high-income countries, the fluctuations in the course of both compared indicators are much greater. It should be noted, however, that the differences in changes in the value of the government-to-private expenditure ratio are much smaller than in the case of the two previous groups of countries (Figure 9) [61,63].

### *3.2. GDP and Total Health Expenditure per Person*

#### 3.2.1. Linear Regression Models

Attempts were made to verify the relationship between GDP and health expenditure per person for individual groups of countries. Initially, three time delays were assumed for the explanatory variable, and three for the dependent variable. Only the best models are presented. The autocorrelation test and heteroscedasticity tests were performed. The residual normality test was not performed due to the small sample size.

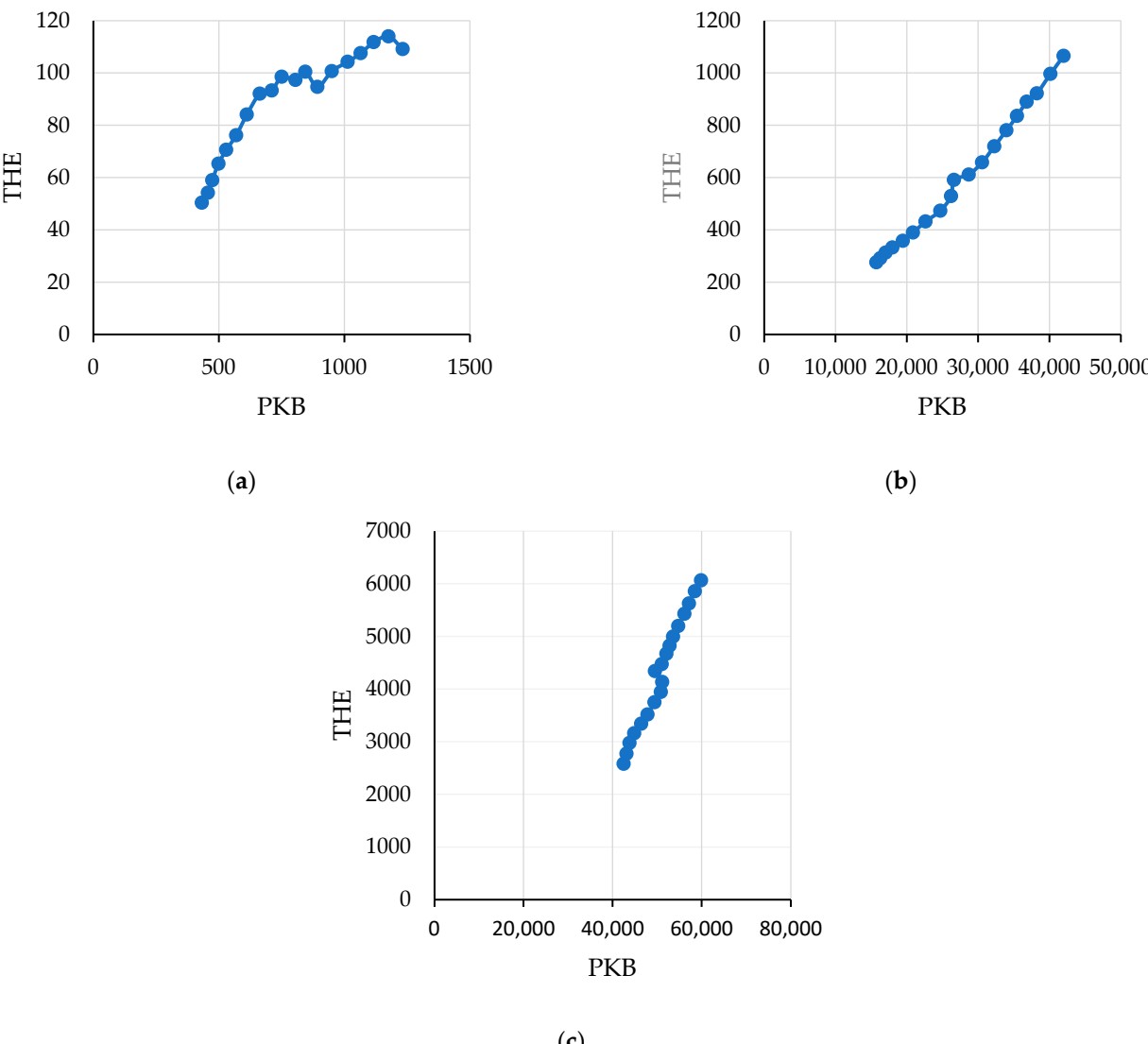

**Figure 7.** Level of GDP PPP (2019 constant prices, in billion international dollars) and total health expenditure (THE) per person PPP (in international dollars): (**a**) Low-income countries (LI); (**b**) Upper-middle income countries (UMI); (**c**) High-income (HI) countries [61,63].

The impact of overall health expenditure on GDP in low-income countries has turned out to occur lagged by one period and with the lag of GDP by one period (Table 1).

**Table 1.** Model 6: KMNK estimation, observations used 2003–2021 (N = 19. Dependent variable (Y): PKB_LI.

| Parameter | Coefficient | Standard Error | T-Student | *p* Value |
|-----------|-------------|----------------|-----------|-----------|
| Const | −7.37015 | 7.79893 | −0.9450 | 0.3587 |
| THE_LI_1 | 0.627341 | 0.190956 | 3.285 | 0.0047 *** |
| GDP_LI_1 | 0.996008 | 0.0148 | 67.11 | <0.0001 *** |

*** Probability value *p* < 0.001.

When attempting to check whether health expenditure in low-income countries was affected by the level of GDP, no satisfactory model was obtained, which would confirm the existence of such a relationship (the only explanatory variable remaining in the model was the first delay of the dependent variable).

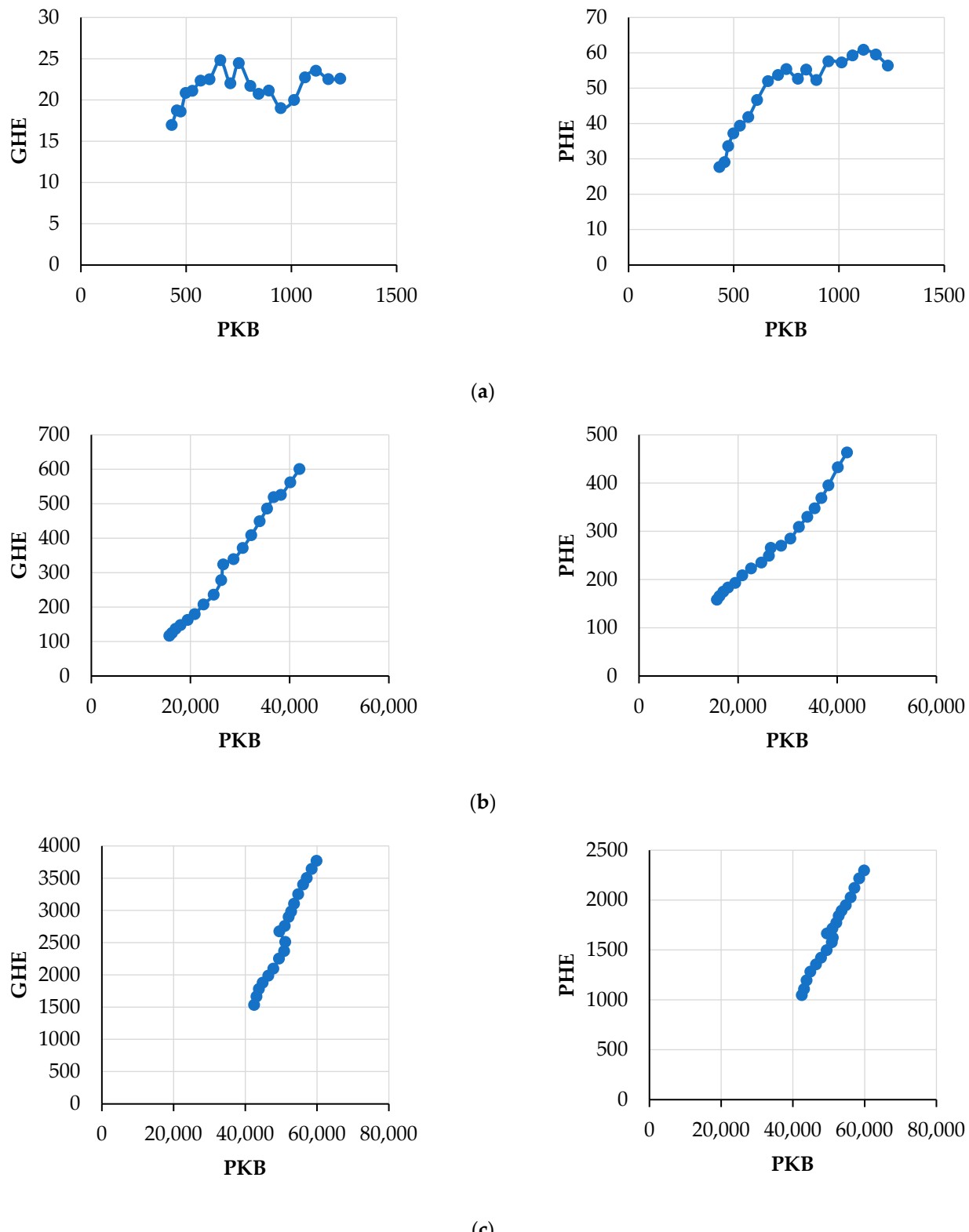

**Figure 8.** Level of GDP PPP (2019 constant prices, in billion international dollars) and health expenditure per person; government (GHE) and private (PHE) (in international PPP dollars): (**a**) Low-income countries (LI); (**b**) Upper-middle income countries (UMI); (**c**) High-income (HI) countries [61,63].

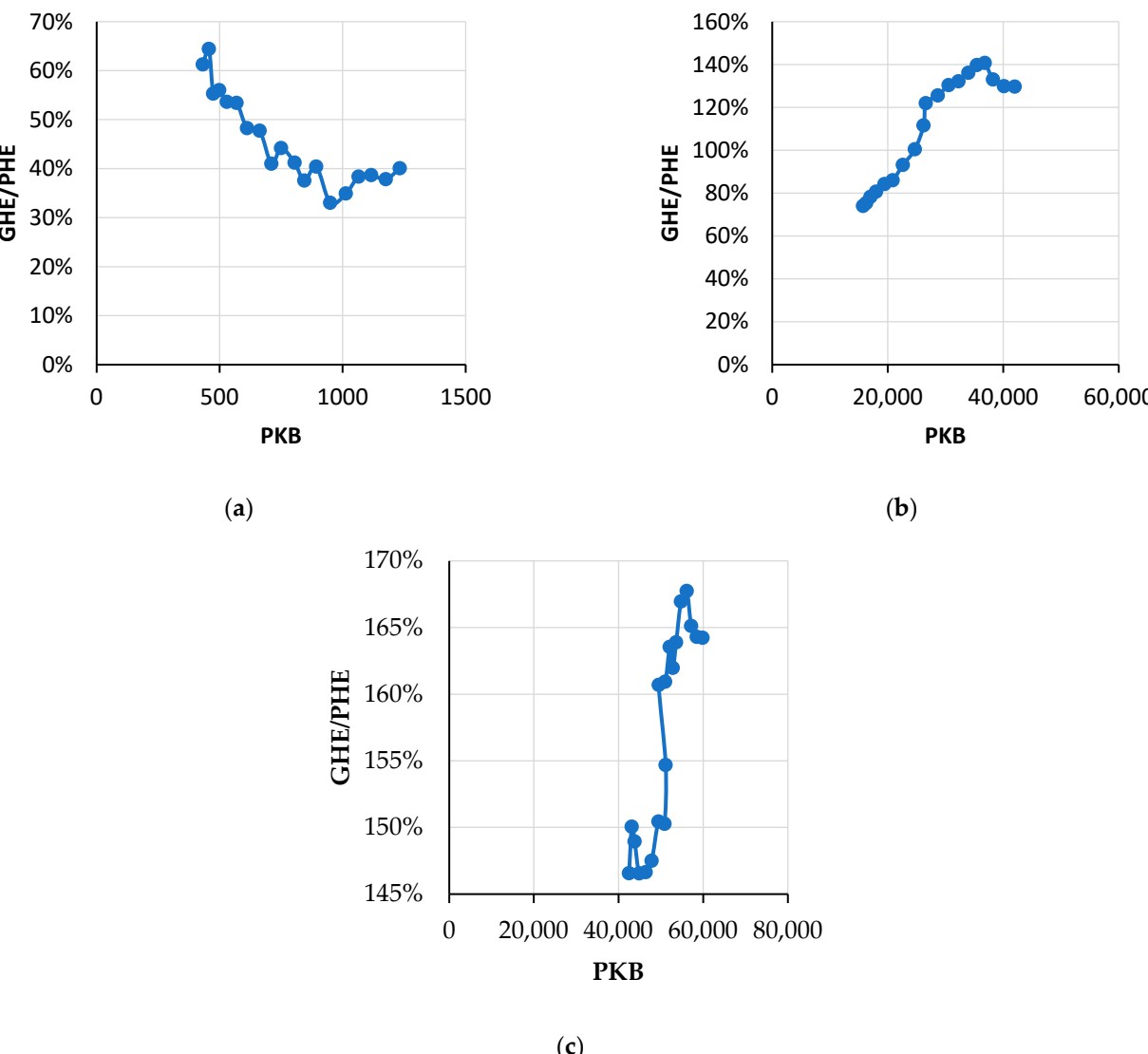

**Figure 9.** Level of GDP PPP (2019 constant prices, in billion international dollars) and the ratio of government to private expenditure on health per person PPP (in international dollars, percentage): (**a**) Low-income countries (LI); (**b**) Upper-middle income countries (UMI); (**c**) High-income (HI) countries [61,63].

In the case of attempting to check whether the GDP level in upper-middle income countries could have been affected by the level of total health expenditure, a satisfactory model was not obtained either.

However, in the case of checking the reverse relation, the possibility of GDP impact was confirmed, but also the first delay in expenditures affected their level (Table 2).

**Table 2.** Model 26: KMNK estimation, observations used 2003–2021 (N = 19). Dependent variable (Y): THE_UMI.

| Parameter | Coefficient | Standard Error | T-Student | *p* Value |
|---|---|---|---|---|
| Const | −129,021 | 20.59 | −6.27 | <0.0001 *** |
| GDP_UMI_1 | 0.02 | 0.002 | 9.2 | <0.0001 *** |
| THE_UMI_3 | 0.43 | 0.08 | 5.34 | 0.0001 *** |

*** Probability value $p < 0.001$.

Similar modelling attempts were made for the high-income group of countries (Table 3). There is a possibility that health expenditure (from the previous period) may have an impact on GDP (and is affected by the GDP of the previous period).

**Table 3.** Model 32: KMNK estimation, observations used 2001–2021 (N = 21). Dependent variable (Y): GDP_HI.

| Parameter | Coefficient | Standard Error | T-Student | *p* Value |
|---|---|---|---|---|
| Const | 11,459.1 | 6434.44 | 1.78 | 0.0939 * |
| THE_HI_1 | 1.77 | 1.02 | 1.73 | 0.1 |
| GDP_HI_1 | 0.644 | 0.21 | 3.049 | 0.0077 *** |

* Probability value $p < 0.05$; *** Probability value $p < 0.001$.

An attempt to construct a model with a reverse relationship was also successful: the first and third delays of the dependent and explanatory variable were adopted (Table 4).

**Table 4.** Model 36: KMNK estimation, observations used 2001–2021 (N = 21). Dependent variable (Y): THE_HI.

| Parameter | Coefficient | Standard Error | T-Student | *p* Value |
|---|---|---|---|---|
| Const | −254.149 | 224.54 | −1.13 | 0.2755 |
| GDP_HI_1 | 0.014 | 0.0073 | 1.913 | 0.0751 * |
| THE_HI_1 | 0.94 | 0.035 | 26.86 | <0.0001 *** |

* Probability value $p < 0.05$; *** Probability value $p < 0.001$.

Summing up, it can be stated that there was no confirmation of the existence of unambiguous, one-way relations between the studied variables in the three analyzed groups of countries (Table 5). Perhaps, in high-income countries, there is a feedback loop between GDP levels and total health spending. By contrast, in low-income countries, GDP appears to depend on health expenditure per capita three years ago. Upper-middle income countries may record the opposite relationship—as their GDP level rises, they can afford to incur ever higher expenditure on health.

**Table 5.** Relations between the studied variables in the three analyzed groups of countries.

| Parameter | GDP Level Depends on Overall Health Expenditure per Person | General Health Expenditure per Person Depends on GDP Level |
|---|---|---|
| Low-Income Countries | Yes—3 periods delay (and with a GDP delay of 1 period) | No |
| Middle-income countries | No | Yes—1 period delay (and with a THE delay of 1 period) |
| High-income countries | Yes—1 period delay (and with a GDP delay of 1 period) | Yes—1 period delay (and with a THE delay of 1 period) |

### 3.2.2. The Causality Study

A Granger causality study was conducted between total health expenditure per person and GDP levels. Delays of three periods were applied. Vector autoregression (VAR) models were constructed. The parameters of the following models were estimated:

$$PKB_t = B_0 + B_1 PKB_{t-1} + B_2 PKB_{t-2} + B_3 PKB_{t-3} + B_4 THE_{t-1} + B_5 THE_{t-2} + B_6 THE_{t-3} \tag{1}$$

$$THE_t = B_0 + B_1 PKB_{t-1} + B_2 PKB_{t-2} + B_3 PKB_{t-3} + B_4 THE_{t-1} + B_5 THE_{t-2} + B_6 THE_{t-3} \tag{2}$$

The null hypotheses for the F-test were as follows:

$$B_4 = B_5 = B_6 = 0 \qquad (3)$$

$$B_1 = B_2 = B_3 = 0 \qquad (4)$$

For low-income countries, the F statistic for Equation (3) was 3.59 and the significance was around 5%, while for Equation (4) it was 0.92 and was negligible at minimum 10% significance level. This means that delays in health expenditure per capita were collectively significant in explaining the variance of GDP, while delays in GDP were not collectively significant in explaining the variance in total health expenditure. This indicates that the variance in overall health expenditure explained the variance of GDP in a Granger sense, but not vice versa, and that overall health expenditure is not an endogenous component of GDP.

For upper-middle income countries, the null hypothesis that health expenditure delays were collectively significant in explaining the variance of GDP should be rejected at a significance level of at least 10%. On the other hand, it should be hypothesized that the delays in GDP were collectively significant in explaining the variance of health expenditure per person at a significance level of less than 1%. This indicates that the variance in GDP resulted in variance in total health expenditure in a Granger sense. This means that, unlike in low-income countries, health expenditure is an endogenous component of GDP in upper-middle income countries.

As can be deducted, the relationship in high-income countries was similar to that in upper-middle income countries, but the conclusions are even stronger. In this group, the null hypothesis that the delays in health expenditure were collectively significant in explaining the variance of GDP should be rejected at the significance level of minimum 10%. On the other hand, it should be hypothesized that the delays in GDP were collectively significant in explaining the variance of health expenditure per person at a significance level of less than 1%. This indicates that the variance in GDP resulted in variance in overall health expenditure in a Granger sense. Hence, this means that, as in high-income countries, health expenditure is an endogenous component of GDP in high-income countries (Table 6).

The summary of Granger's causality analysis produces similar conclusions as in the case of regression models, except in one case. It should be remembered that the studied hypothesis about the total significance of all three delays was tested, not each individual one of them. Hence, considering delays not by one (first) period, but by all three, the hypothesis that GDP does depend on total health expenditure in developed countries should be rejected (Table 7).

Taking these explanations into account, it means that consistent conclusions were obtained using both methods.

Spending on health is an endogenous component of GDP in high- and upper-middle income countries, but it is not in low-income countries.

These relations might be visualized on charts. The charts below show the impulse responses over the 10-year forecast horizon (Figures 10–12).

It is quite clear that health expenditure can affect GDP in low-income countries.

Projections also suggest that the two factors may affect each other in upper-middle income countries. This means that at this level of development, both GDP growth causes an increase in health expenditure, and vice versa. Moreover, an increase in the level of GDP may cause it to rise in the future; this may also be the case for health expenditure (at least two years after the initial impulse).

In the case of high-income countries, on the other hand, GDP has an impact on future health expenditure per person (this impact grows in the first four years from the GDP increase and expires in subsequent years).

**Table 6.** Causality test: GDP and general health expenditure per person.

| Test | Parameter | LI | UMI | HI |
|---|---|---|---|---|
| Test GDP | $GDP_{t-1}$ | 0.932099 *** (0.250011) | 1.36169 *** 0.328211 | 0.970287 ** 0.313632 |
| | $GDP_{t-2}$ | −0.327551 (0.329595) | −1.19068 0.685477 | −0.707000 0.686058 |
| | $GDP_{t-3}$ | 0.250404 (0.231068) | −0.368404 0.724116 | 0.130192 0.657163 |
| | $PHE_{t-1}$ | 0.858237 (0.559331) | 56.9031 ** 22.8999 | 10.0673 13.3101 |
| | $PHE_{t-2}$ | −0.894814 (0.763696) | −5.44681 31.1131 | −6.18442 19.9357 |
| | $PHE_{t-3}$ | 1.93868 ** (0.705971) | −13.2966 15.4559 | −1.03001 12.6678 |
| | C | 20.2779 (16.3475) | 9063.15 * 4971.54 | 15,892.2 15,993.8 |
| | $R^2$ | 0.999235 | 0.997732 | 0.978864 |
| | F test | 3.5953 (0.0539) | 2.5662 (0.1131) | 0.30811 (0.8191) |
| Test THE | $GDP_{t-1}$ | 0.163567 (0.131553) | 0.0142389 *** 0.00444608 | 0.0428748 *** 0.00550212 |
| | $GDP_{t-2}$ | −0.285789 (0.173429) | 0.00642896 0.00928577 | −0.00581396 0.0120357 |
| | $GDP_{t-3}$ | 0.122716 (0.121585) | −0.000179524 0.00980919 | 0.0278481 ** 0.0115288 |
| | $PHE_{t-1}$ | 0.980202 (0.294314) | 0.345213 0.310211 | 0.652974 ** 0.233503 |
| | $PHE_{t-2}$ | −0.0796538 (0.401848) | −0.139543 0.421472 | −0.323966 0.349737 |
| | $PHE_{t-3}$ | 0.0607657 (0.371474) | 0.163584 0.209372 | 0.344009 0.222235 |
| | C | 4.39725 (8.60187) | −150.349 ** 67.3466 | −1460.15 *** 280.583 |
| | $R^2$ | 0.985939 | 0.999486 | 0.999842 |
| Test F | | 0.92027 [0.4659] | 14.420 [0.0006] | 27.316 [0.0000] |

* Probability value $p < 0.05$; ** Probability value $p < 0.01$; *** Probability value $p < 0.001$.

**Table 7.** Causal effect relations between GDP and total health expenditure per person by country group.

| Parameter | The Level of GDP Depends on Overall Health Expenditure per Person | Total Health Expenditure per Person Depends on the Level of GDP |
|---|---|---|
| Low Income countries | Yes | No |
| Upper-middle income countries | No | Yes |
| High-income countries | No | Yes |

3.2.3. GDP vs. Private and Government Spending on Health per Person

A similar analysis will be performed below for private health expenditure per person (PHE). It will begin with a Granger causality study.

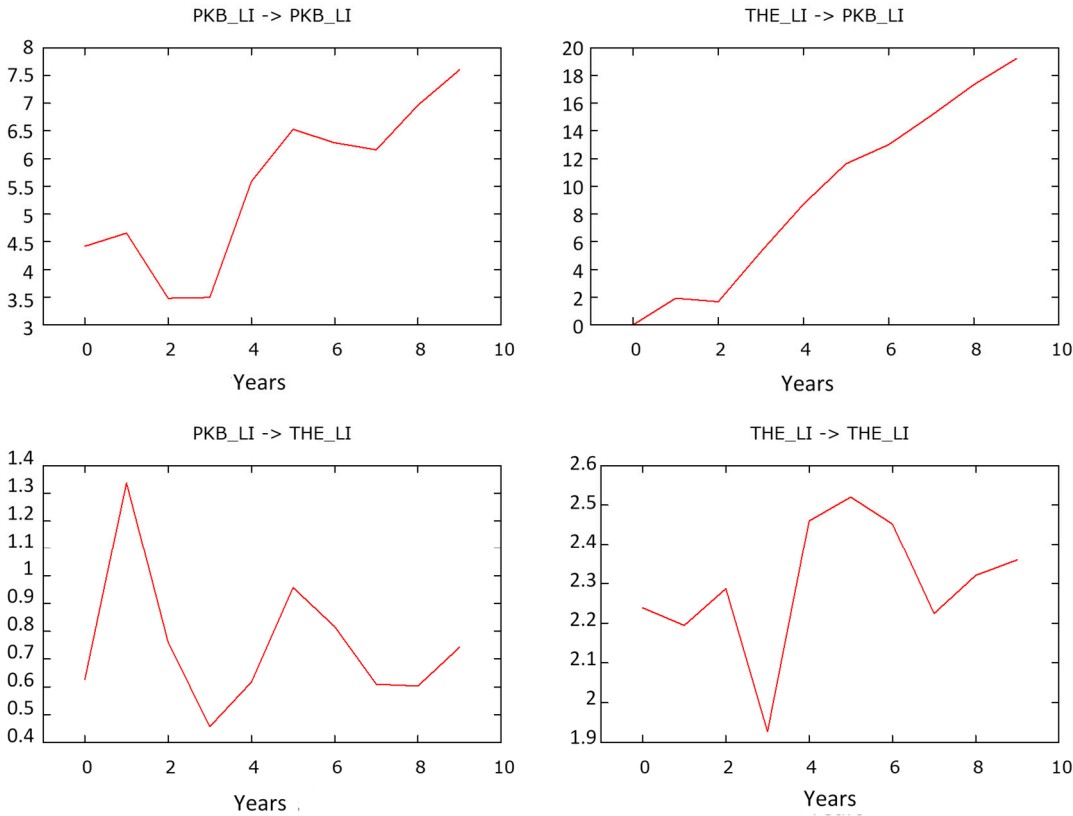

**Figure 10.** Dependencies between GDP and THE in low-income countries.

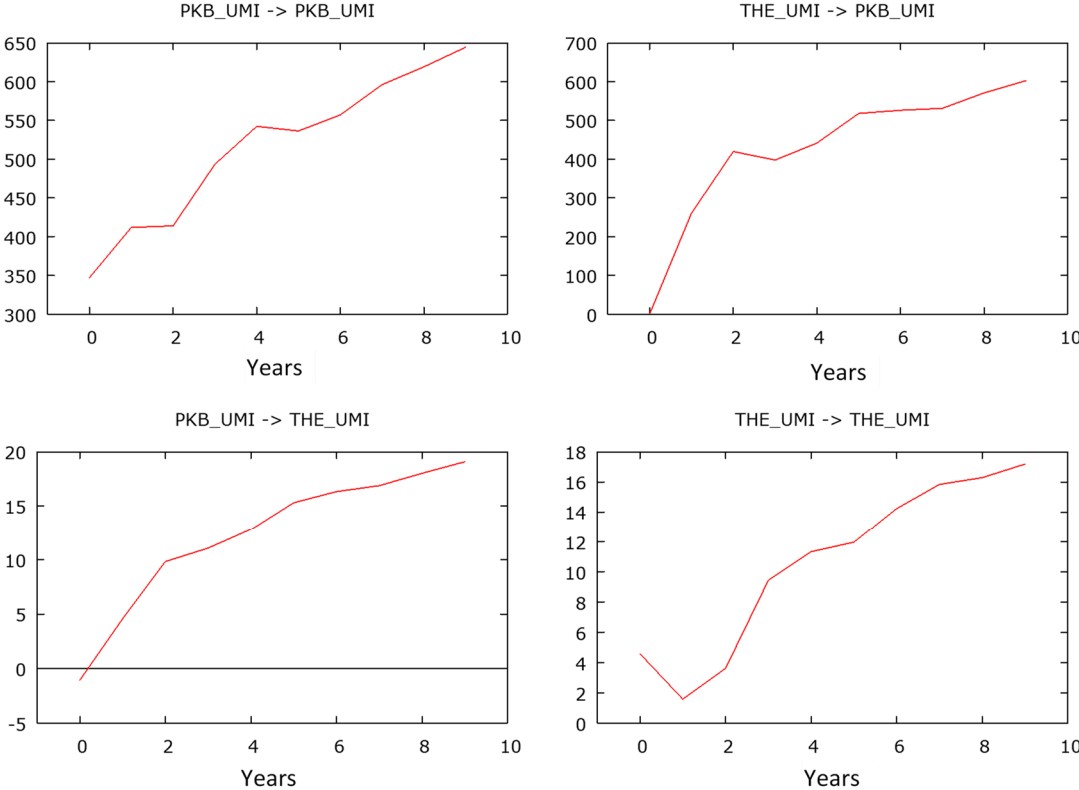

**Figure 11.** Relationships between GDP and THE in upper-middle income countries.

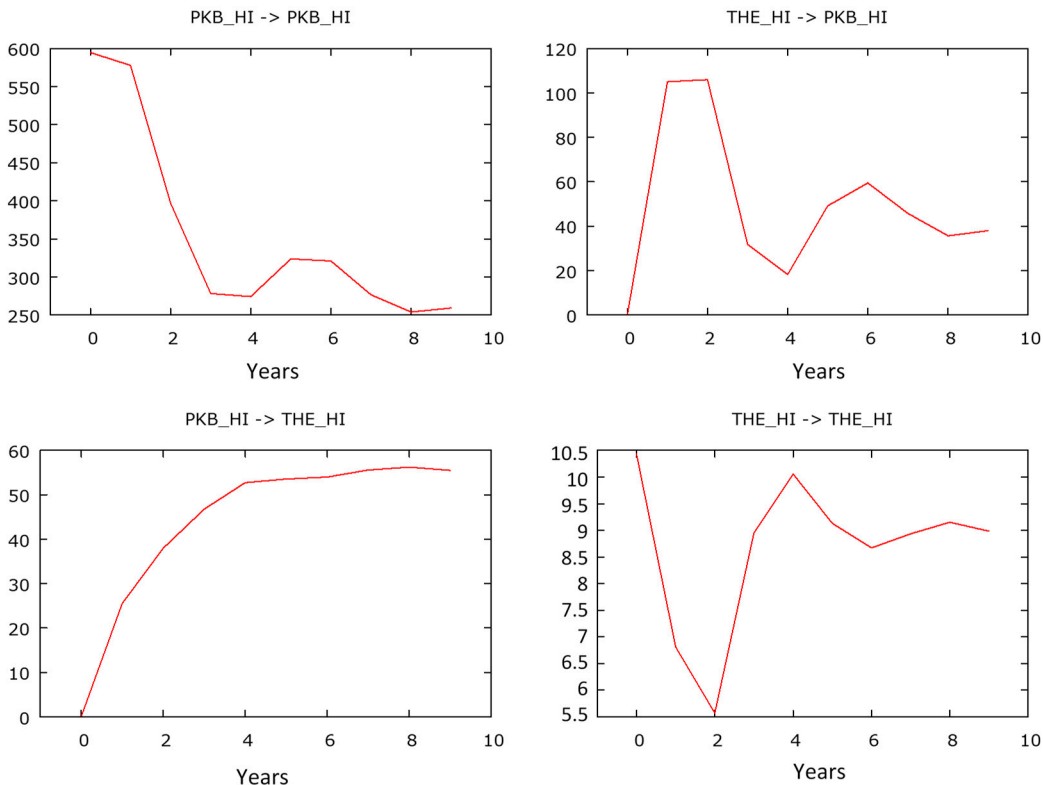

**Figure 12.** Relationships between GDP and THE in high income countries.

The parameters of the following models were estimated:

In the case of low-income countries, there is no reason to reject any of the null hypotheses (3 and 4). Thus, neither did private expenditure on health per person explain the level of GDP, nor was the level of GDP the cause of private expenditure on health, in Granger's sense.

Similar calculations were made for upper-middle income countries. The null hypothesis (3) should be rejected when there is a possible impact of private health expenditure on GDP at the 1% significance level. On the other hand, the null hypothesis (4) about the occurrence of a reverse relation should be accepted. This means that GDP had no impact on the level of private health spending in UMI countries, while PHE had an impact on the level of GDP. This means that health expenditure is an endogenous component of GDP in upper-middle income countries.

In the case of developed countries, none of the null hypotheses was rejected, i.e., neither GDP was the cause of PHE nor PHE was the cause of GDP.

Detailed results are summarized in the table below (Table 8).

A similar analysis is performed below for government health expenditure per person (GHE).

The results indicate that government spending on health affected the GDP of low-income countries, in a Granger sense. Equation (3) was rejected at the 5% significance level, and Equation (4) was not rejected at a min. 10% significance level.

Moreover, the level of GDP significantly influenced government spending per capita on health in both groups of the richer countries (Table 9).

### 3.3. The Impact of the Pandemic and the War in Ukraine on Energy and GDP

On 24 February 2022, the Russian Federation began its invasion of Ukraine, on which the whole world is now focused [61]. In the shadow of these tragic events, the climate crisis caused by the burning of fossil fuels, leading to anthropogenic emissions of carbon dioxide into the atmosphere, is worsening [62]. The war has given rise to a number of fundamental questions and answers. The most important question is whether renewable energies (RES),

energy efficiency and nuclear energy can save the climate, while at the same time protecting the EU and the world from a serious crisis caused by the disruption of supplies, or the closure of supplies, from Russia.

**Table 8.** Causality test: GDP and private health expenditure per person.

| Test | Parameter | LI | UMI | HI |
|---|---|---|---|---|
| Test GDP | $GDP_{t-1}$ | 0.770417 ** 0.309805 | 0.561125 0.322382 | 0.807732 * 0.367505 |
| | $GDP_{t-2}$ | −0.316687 0.376960 | −0.163079 0.382548 | −0.581889 0.433460 |
| | $GDP_{t-3}$ | 0.318247 0.274244 | 0.00198330 0.237233 | 0.0863834 0.325576 |
| | $THE_{t-1}$ | 1.74835 1.21634 | 26.9171 19.0709 | 5.09995 20.6595 |
| | $THE_{t-2}$ | −0.383503 1.56290 | 34.4204 30.7256 | 22.2953 30.4690 |
| | $THE_{t-3}$ | 3.29692 * 1.78834 | −1.63888 25.2819 | −18.5647 17.9119 |
| | C | 21.4033 20.4900 | 3038.39 ** 1139.48 | 17,655.4 10,070.5 |
| | $R^2$ | 0.999004 | 0.998677 | 0.981282 |
| | Test F | 1.9871 (0.1799) | 6.7774 (0.0090) | 0.77869 (0.5323) |
| Test THE | $GDP_{t-1}$ | 0.0951862 0.0811435 | 0.00274435 0.00471767 | 0.0102348 0.00646679 |
| | $GDP_{t-2}$ | −0.163821 0.0987326 | −0.00375898 0.00559813 | −0.00753511 0.00762737 |
| | $GDP_{t-3}$ | 0.0674311 0.0718293 | −0.000151511 0.00347162 | 0.00669497 0.00572899 |
| | $THE_{t-1}$ | 1.04106 *** 0.318580 | 1.02822 *** 0.279080 | 1.23318 0.363534 |
| | $THE_{t-2}$ | −0.102144 0.409351 | −0.580330 0.449632 | −0.557732 0.536148 |
| | $THE_{t-3}$ | 0.0390989 0.468397 | 0.720417 * 0.369970 | 0.183253 0.315186 |
| | C | 2.50898 5.36670 | 14.6788 16.6749 | −166.548 177.206 |
| | $R^2$ | 0.985827 | 0.997665 | 0.998957 |
| | Test F | 0.93770 [0.4583] | 0.59805 [0.6307] | 1.1514 [0.3756] |

* Probability value $p < 0.05$; ** Probability value $p < 0.01$; *** Probability value $p < 0.001$.

The outbreak of war in the east has once again shown how important the issue of energy security is. Both renewable energy and nuclear energy are necessary not only for the climate, but also for independence from Russia and the problems associated with the disruption of fossil fuel supplies. The role of renewables and nuclear power will undoubtedly be crucial in the coming decades. The European Union's current climate policy needs to be reviewed in the light of Russia's aggression against Ukraine. It is therefore necessary to build an energy mix based on a variety of energy sources, both renewable and coal.

**Table 9.** Causality test: GDP and government health expenditure per capita.

| Test | Parameter | LI | UMI | HI |
|---|---|---|---|---|
| Test GDP | $GDP_{t-1}$ | 1.10345 *** <br> 0.217185 | 1.12064 *** <br> 0.341270 | 1.09183 ** <br> 0.349779 |
| | $GDP_{t-2}$ | −0.385976 <br> 0.301832 | −0.619691 <br> 0.739063 | −0.717498 <br> 0.688672 |
| | $GDP_{t-3}$ | 0.229849 <br> 0.202753 | 1.26896 * <br> 0.580056 | 0.465000 <br> 0.563948 |
| | $GHE_{t-1}$ | 1.56433 <br> 0.867990 | −4.73946 <br> 36.6117 | 7.44400 <br> 13.4217 |
| | $GHE_{t-2}$ | −2.42166 * <br> 1.15820 | −59.5493 <br> 44.5117 | −12.9369 <br> 18.0796 |
| | $GHE_{t-3}$ | 3.42224 *** <br> 0.987437 | 28.6698 <br> 22.7762 | 6.45631 <br> 11.0408 |
| | C | 11.0374 <br> 12.5589 | −6316.87 * <br> 3480.55 | 6019.19 <br> 10,368.4 |
| | $R^2$ | 0.999372 | 0.997449 | 0.978510 |
| | Test F | 5.0969 <br> (0.0214) | 1.9104 <br> (0.1919) | 0.24823 <br> (0.8608) |
| Test THE | $GDP_{t-1}$ | 0.0871543 <br> 0.0739304 | 0.00763302 ** <br> 0.00259542 | 0.0236094 ** <br> 0.00779372 |
| | $GDP_{t-2}$ | −0.130487 <br> 0.102744 | 0.00739233 <br> 0.00562072 | 0.00288453 <br> 0.0153449 |
| | $GDP_{t-3}$ | 0.0522872 <br> 0.0690175 | −0.0106850 ** <br> 0.00441144 | 0.0174740 <br> 0.0125658 |
| | $GHE_{t-1}$ | 0.833610 ** <br> 0.295466 | 0.496350 <br> 0.278439 | 0.547820 * <br> 0.299059 |
| | $GHE_{t-2}$ | −0.166732 <br> 0.394253 | 0.602323 <br> 0.338521 | −0.147387 <br> 0.402847 |
| | $GHE_{t-3}$ | 0.126234 <br> 0.336126 | −0.316463 * <br> 0.173218 | 0.257000 <br> 0.246009 |
| | C | 0.379131 <br> 4.27509 | −46.6398 <br> 26.4703 | −1046.53 *** <br> 231.028 |
| | $R^2$ | 0.974656 | 0.999519 | 0.999355 |
| Test F | | 0.68797 <br> (0.5797) | 24.601 <br> (0.0001) | 14.313 <br> (0.0006) |

* Probability value $p < 0.05$; ** Probability value $p < 0.01$; *** Probability value $p < 0.001$.

The Research Center for Energy and Clean Air has prepared a detailed report on the amount of raw material that Russia has sold to other countries since the beginning of the war in Ukraine. Data show that since the beginning of the war in Ukraine, EUR 63 billion of fossil fuels have been exported from Russia. EU countries imported a total of EUR 44 billion. The largest importers were Germany (9.1 billion), Italy (6.9 billion), China (6.7 billion), the Netherlands (5.6 billion), Turkey (4.1 billion) and France (3.8 billion). Germany paid EUR 9.1 billion for Russian fossil fuels, EUR 6.4 billion for gas pipelines, EUR 2 billion for oil, EUR 644 million for petroleum products and EUR 92.6 million for coal [63]. According to CREA calculations, this expenditure has been around EUR 150 million per day since the beginning of the war.

The most important supplier of fossil fuels to Poland is Russia. Their share of imports was 87% for oil, 72% for natural gas and 62% for hard coal (average of 20 years). In the atypical pandemic years of 2021 and 2020, the purchase of energy raw materials increased

significantly, despite decreasing volumes. Imports of each individual energy commodity have increased significantly over the past 20 years. Carbon imports increased most by up to 750% from a small volume of 1.5 million tons in 2000 to 12.9 million tons in 2020 (where energy demand was lower than in previous years due to the coronavirus pandemic). We import twice as much natural gas today as in 2000, and oil imports have increased by 41% in 20 years. After Russia's invasion of Ukraine, Poland's GDP will fall by 0.2–3.3% by the end of 2022, depending on the extent of the rise in commodity prices [64].

Oil deliveries from Russia to foreign ports decreased by 20% in the first three weeks of April 2022, compared to the period from January to February 2022 (before the invasion of Ukraine). There has been a significant increase in oil deliveries from Russia to India and Egypt. However, deliveries to these new destinations are not sufficient to compensate for the decline in exports to Europe [65].

In the short term, Russia cannot replace Europe as a source of demand. Most of Russia's fossil fuel exports are transported to Europe via pipelines and port roads in the Baltic and Black Seas. LNG terminals, or pipelines through which gas exports could be diverted, simply do not exist [66].

Ukraine imported 75% of its fuel from Russia and Belarus. More than 10% of the fuel needs were covered by PKN Orlen's refinery in Możyki, Lithuania, via Belarus. After the invasion of Ukraine, Russia and Belarus cut off fuel supplies. Ukraine currently has six refineries, but they are very outdated. Fuel production takes place exclusively at the Krzemenchuk plant in the central part of Ukraine, in the Poltava region, and at the small Szebelinka plant. Both institutions covered more than 12 percent of the country's needs [34].

According to Dragon Capital, a Ukrainian investment company, the decline in GDP in Ukraine amounted to 45 percent in March 2022. Dragon Capital estimates that Ukraine's budget deficit could reach 26% of GDP by the end of 2022. The main reason for this situation is the blockade and demining of the Black Sea by Russian vessels, which severely limits Ukraine's export and import capacities. In addition, the economic situation of the country was influenced by the fact that almost five million Ukrainians fled from the war across the border of Ukraine [67].

As the Statista.com infographic, based on data from the Institute for the World Economy in Kiel, shows, relatively speaking, no country came any closer to Estonia's participation in the first four weeks of the war. The contribution of one of the three Baltic States that were once part of the former Soviet Union amounted to EUR 0.22 billion in military aid. This corresponds to 0.8% of the country's GDP. Up to that date, US financial assistance amounted to 0.04% of US economic output. In this comparison, Poland ranks second, with aid amounting to 0. 18% of GDP [68].

According to the World Bank's Office of Chief Economist for Europe and Central Asia, "War in the Region", the war has caused a huge economic shock by hampering recovery from the COVID-19 pandemic. Extreme poverty is increasing significantly. The report focuses on a region comprising 50 countries. For the entire region of Europe and Central Asia, the report estimates that the economy will shrink by 4.1% in 2022, twice as much as during the pandemic recession. The neighboring countries of Russia and Ukraine are particularly affected because of their trade, financial and migration links with these countries. Russia is the main exporter of energy and metals, including iron, aluminum and palladium, and together with Ukraine it accounts for more than 15 percent of global wheat exports. Supply shortages and rising energy and food prices are contributing significantly to the rise in inflation in the region [69].

After Russia's invasion of Ukraine, coal increased by 97%, oil (30%), electricity and gas (45%), metals (18%) and agricultural products (32%) [70].

Currently, electricity generation in Ukraine runs on remnants of nuclear power plants (the largest in Zaporozhye with an output of 6 GW was taken over by Russian troops). Nuclear energy is only half-functioning, as is hydropower, a decline to about 3 GW, while thermal power plants—most of which are either destroyed or shut down—operate about 3 GW. Both in the direct war zone (grids and power plants) as well as at the switching

points in the hinterland and in the western part of Ukraine, great destruction has been recorded. It must be emphasized that in Ukraine most rail transport was electrified and there was a radical reduction in troop and supply capacities [71,72].

Europe's dependence on Russian gas and oil in the face of the war in Ukraine has become one of the European Union's biggest problems. Phasing out fossil fuels and investing in alternative energy sources are challenges not only for Europe but also for global economies. The aggression of Russia, one of the world's main suppliers of hydrocarbons, against Ukraine, makes it necessary to review the energy policies of many countries, including the objectives of the energy transition [73].

The war also poses challenges (and costs) to the health care of countries that have decided to help Ukrainian refugees. The massive influx of refugees from Ukraine poses additional challenges to the health system, already strained by the COVID-19 pandemic and staffing shortages (e.g., in Poland). During the first 12 days of the war, 1.7 million people left Ukraine, of whom almost 1.1 million came to Poland (mostly women and children). There is an urgent need for the health system to respond to the massive need for mental health professionals, especially those working with children who may need psychological or psychiatric support, including post-traumatic stress disorder. Experts point out that primary health care will bear the brunt of long-term benefits [74].

Another challenge is the low vaccination rate of Ukrainians against COVID-19, which amounts to 34.5 percent. Refugees from Ukraine received free COVID-19 tests and access to vaccinations against COVID-19.

## 4. Conclusions

The directions of causality (in Granger's sense) are summarized in the table below. It indicates that from the point of view on the sources of funding health expenditure, it can be concluded that government expenditure affects the level of GDP in the case of low-income countries, while private expenditure, which influences GDP, gains more importance in the upper-middle income countries. By contrast, the level of GDP has an impact on government health expenditure in the upper-middle-income and high-income groups, but not in the low-income countries (Table 10).

**Table 10.** Cause-and-effect relationships between GDP and private and government health expenditure per capita by country groups.

| Parameter | The Level of GDP Depends on Health Expenditure per Person | | Health Expenditure per Person Depends on the Level of GDP | |
|---|---|---|---|---|
| | Private Expenditure | Government Expenditure | Private Expenditure | Government Expenditure |
| Low-Income Countries | No | Yes | No | No |
| Upper-middle income countries | Yes | No | No | Yes |
| High-income countries | No | No | No | Yes |

Although the EU economy and its society are still struggling with the greatest experience since the Second World War, the COVID-19 pandemic, it has strengthened the region before the next crisis, such as the war in Ukraine.

Russia's invasion of Ukraine in 2022 is predicted to reduce the EU's GDP by up to 1. 8 percent, leading to an increase in the unemployment rate of 3.7 million people and inflation (between 1.3 and 3 percent), compared to pre-war levels.

The pandemic has led to a sharp rise in unemployment, particularly in Mediterranean countries. This has been avoided by the so-called new EU countries. In the event of a war in Ukraine, unlike Spain, Italy or Greece, the new EU countries could be affected by the crisis.

The war in Ukraine forced the countries of the European Union to change their energy supply policies. The aggression of Russia, one of the world's main suppliers of hydrocarbons, against Ukraine, makes it necessary to review the energy policies of many countries, including the objectives of the energy transition. It has a very strong impact on energy prices and there is a risk that the conflict will slow down the phase-out of coal in some countries of the European Union.

With respect to the impact of the war on food prices, it should be emphasized that Ukraine and Russia account for 25% of world grain exports. If Russia is unable to export grain and Ukraine is unable to produce and export grain, it will have a major destabilizing effect on markets. The arrival of refugees from Ukraine, in terms of the health system, causes the need for an increased number of professionals, especially for the mental health of children, and an increased number of vaccinations against COVID-19.

Poland has received the largest number of refugees from Ukraine so far. The Polish health system is in urgent need of systemic solutions, prepared by both the state and local authorities, to move from spontaneous measures to the implementation of legal measures.

Since 24 February 2022, the day Russia attacked Ukraine, the health protection of countries that have taken in refugees has faced another unique challenge, namely the admission of patients at the eastern border. The aim remains the same: to ensure the performance of the entire health system in emergency situations. On the one hand, health facilities should be adequately prepared to provide medical assistance to refugees from war-torn Ukraine. On the other hand, our healthcare system must ensure that all patients in need have access to diagnosis and treatment. People fleeing Ukraine should have access to a wide range of health services, and to the healthcare system in the country where they are staying.

We are witnessing one of the biggest, if not the biggest, humanitarian crises in Europe since the Second World War. The escalation is causing more civilian casualties every day.

Ukraine's medical care system has collapsed. Hospitals and other health centers are bombed. The challenge for the Ukrainian health care system is not only the lack of basic equipment and medicines, but also the medical transport that allows the transport of injured persons to hospitals in neighboring countries.

The war also has an impact on the pharmaceutical market. Although there are currently no reports of disruptions in the continuity of pharmaceutical production, or of the lack of supplies to wholesalers and pharmacies, this does not mean that there will be no problems in the near future.

In addition, the Russian invasion of Ukraine is causing disruptions in global markets. Since the beginning of the conflict, world market prices for key raw materials, in particular fuels, have risen sharply. Russian attacks on cultivation and transport infrastructure have limited Ukraine's ability to export agricultural and food products, further exacerbating the global food crisis.

The constant observation and analysis of the effects of the war and its consequences, not only in Ukraine, but also in Europe and the world, forces scientists to intensify their research.

In further scientific publications, the author will develop research related to health expenditures related to the war in Ukraine. In this context, it will also examine the development of the pharmaceutical market.

**Funding:** This research was funded by the National Science Center Poland (project No. 2013/09/N/HS4/03672).

**Institutional Review Board Statement:** Not applicable.

**Informed Consent Statement:** Not applicable.

**Data Availability Statement:** All data are presented in this article. Data sharing is not applicable to this article.

**Conflicts of Interest:** The author declares no conflict of interest.

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
