# Peer review of "Health Care Financing and Economic Performance during the Coronavirus Pandemic, the War in Ukraine and the Energy Transition Attempt"

_sustainability, doi:10.3390/su141710601_

Round 1

Reviewer 1 Report

Manuscript review: Health care financing and economic performance during the coronavirus pandemic, the war in Ukraine and the energy transition attempt  (sustainability-1839759).

In the manuscript, the author examines the effects of covid-19, the war in Ukraine, the increase in energy expenditure as a function of GDP, and financial expenditure on health. The analysis of the manuscript is multidimensional and covers the period 2020-2021. The manuscript is very interesting and represents an original presentation of the subject, which is commendable. Of particular note is the very solid literature review that the author has made in the manuscript. In my opinion, improvements should be made:

1.       Literature sources are missing in Figures 1-9.

2.       Section 3. 3 (line 376) should, in my opinion, be expanded to highlight the whole research problem.

3.       Conclusions should be developed and research results in the field of energy highlighted

Besides, I have no comments. I think the article is published.

Author Response

Dear Reviewer

Thank you for your valuable comments and comments. I have included them all in the text of the manuscript. I marked the changes green.

Best regards,

Kornelia Piech

  1. Literature sources are missing in Figures 1-9.

Done. Marked in green. At the suggestion of the reviewer, literature was added to Figures 1 to 9.

OECD Data. Health spending. Available online: https://data.oecd.org/healthres/health-spending.htm (accessed on 16 August 2022).

Current health expenditure per capita (current US$) - OECD members. Available online: https://data.worldbank.org/indicator/SH.XPD.CHEX.PC.CD?locations=OE (accessed on 16 August 2022).

OECD Data. Health expenditure and financing. Available online: https://stats.oecd.org/index.aspx?DataSetCode=SHA (accessed on 16 August 2022).

OECD Data. Health expenditure in relation to GDP. Available online: https://www.oecd-ilibrary.org/sites/e26f669c-en/index.html?itemId=/content/component/e26f669c-en (accessed on 16 August 2022).

OECD Data. Health Expenditure. Available online: https://www.oecd.org/els/health-systems/health-expenditure.htm (accessed on 16 August 2022).

  1. Section 3. 3 (line 376) should, in my opinion, be expanded to highlight the whole research problem.

Done. Marked in green.

Currently, electricity generation in Ukraine runs on remnants of nuclear power plants (the largest in Zaporozhye with an output of 6 GW was taken over by Russian troops). Nuclear energy is only half-functioning, as is hydropower – a decline to about 3 GW, while thermal power plants – most of which are either destroyed or shut down – operate about 3 GW. Both in the direct war zone (grids and power plants) as well as at the switching points in the hinterland and in the western part of Ukraine, great destruction has been recorded. It must be emphasized that in Ukraine most rail transport was electrified and there was a radical reduction in troop and supply capacities.

Europe’s dependence on Russian gas and oil in the face of the war in Ukraine has become one of the European Union’s biggest problems. Phasing out fossil fuels and investing in alternative energy sources are challenges not only for Europe but also for global economies. The aggression of Russia, one of the world’s main suppliers of hydrocarbons, against Ukraine, makes it necessary to review the energy policies of many countries, including the objectives of the energy transition.

The war also poses challenges (and costs) to the health care of countries that have decided to help Ukrainian refugees. The massive influx of refugees from Ukraine poses additional challenges to the health system, already strained by the COVID-19 pandemic and staffing shortages (e.g. in Poland). During the first 12 days of the war, 1.7 million people left Ukraine, of whom almost 1.1 million came to Poland (mostly women and children). There is an urgent need for the health system to respond to the massive need for mental health professionals, especially those working with children who may need psychological or psychiatric support, including post-traumatic stress disorder. Experts point out that primary health care will bear the brunt of long-term benefits.

Another challenge is the low vaccination rate of Ukrainians against COVID-19, which amounts to 34.5 percent. Refugees from Ukraine received free COVID-19 tests and access to vaccinations against COVID-19.

  1. Conclusions should be developed and research results in the field of energy highlighted. Marked in green.

The war in Ukraine forced the countries of the European Union to change their energy supply policies. The aggression of Russia, one of the world’s main suppliers of hydrocarbons, against Ukraine, makes it necessary to review the energy policies of many countries, including the objectives of the energy transition. It has a very strong impact on energy prices and there is a risk that the conflict will slow down the phase-out of coal in some countries of the European Union.

The impact of the war on food prices should be emphasized Ukraine and Russia account for 25% of world grain exports. If Russia is unable to export grain and Ukraine is unable to produce and export grain, it will have a major destabilizing effect on markets. The arrival of refugees from Ukraine in terms of the health system causes the need for an increased number of professionals, especially on the mental health of children, an increased number of vaccinations against COVID-19.

Poland has received the largest number of refugees from Ukraine so far. The Polish health system is in urgent need of systemic solutions prepared by both the state and local authorities to move from spontaneous measures to implementation of legal measures.

Reviewer 2 Report

The article entitled "Health care financing and economic performance during the coronavirus pandemic, the war in Ukraine and the energy" transition attempt" is very well written by the authors. The research design and presentation is very well organized. However, introduction part is long and some repetition of the same content. please rewrite this part.

In the abstract this sentence " Therefore, this article attempts to analyze the relationship between these values" is not clear, must be elaborate.

Presentation of the data and tables are very clear. Conclusion is well written and informative. 

Therefor, I strongly, recommend this article to publish in Sustainability journal after correcting above minor comments.

Author Response

Dear Reviewer

Thank you for this review. I have considered all your suggestions (and the suggestions of other reviewers have also been considered). I marked the changes green.

Best regards,

Kornelia Piech

The article entitled "Health care financing and economic performance during the coronavirus pandemic, the war in Ukraine and the energy" transition attempt" is very well written by the authors. The research design and presentation is very well organized. However, introduction part is long and some repetition of the same content. please rewrite this part. Done.

In the abstract this sentence "Therefore, this article attempts to analyze the relationship between these values" is not clear, must be elaborate. Done. Marked in green.

The way in which health care is financed has a significant impact both on the state of health of the population and on the level of financial resources allocated to health care (e.g. health contributions according to income).

Presentation of the data and tables are very clear. Conclusion is well written and informative. Therefor, I strongly, recommend this article to publish in Sustainability journal after correcting above minor comments.
